# Stability of Mycotoxins in Individual Stock and Multi-Analyte Standard Solutions

**DOI:** 10.3390/toxins12020094

**Published:** 2020-01-30

**Authors:** Mariya Kiseleva, Zakhar Chalyy, Irina Sedova, Ilya Aksenov

**Affiliations:** Federal Research Centre of Nutrition and Biotechnology, Ust’inskiy pr., 2/14, 109240 Moscow, Russian; brew@ion.ru (Z.C.); isedova@ion.ru (I.S.); aksenov@ion.ru (I.A.)

**Keywords:** mycotoxins, stability, multi-mycotoxin detection, standard solution, UV-spectroscopy, HPLC-MS/MS

## Abstract

Standard solutions of mycotoxins prepared in RP HPLC solvents from neat standards are usually used for analytical method development. Multi-mycotoxin HPLC-MS/MS methods necessitate stability estimation for the wide spectrum of fungal metabolites. The stability of individual diluted stock standard solutions of mycotoxins in RP-HPLC solvents and multi-analyte HPLC-MS/MS calibrants was evaluated under standard storage and analysis conditions. Individual stock standard solutions of aflatoxins, sterigmatocystin, A- and B-trichothecenes, zearalenone and its analogues, ochratoxin A, fumonisins, *Alternaria* toxins, enniatins and beauvericin, moniliformin, citrinin, mycophenolic, cyclopiazonic acids and citreoviridin were prepared in RP-HPLC solvents and stored at −18 °C for 14 months. UV-spectroscopy was utilized to monitor the stability of analytes, excluding fumonisins. The gradual degradation of α-, β-zearalenol and α-, β-zearalanol in acetonitrile was detected. Aflatoxins and sterigmatocystin, zearalenone, *Alternaria* toxins, enniatins and beauvericin, citrinin, mycophenolic, cyclopiazonic acids and citreoviridin can be referred to as stable. The concentration of the majority of trichothecenes should be monitored. Diluted multi-mycotoxin standard in water/methanol (50/50 *v/v*) solutions acidified with 0.1% formic acid proved to be stable in silanized glass at 23 °C exposed to light for at least 75 h (CV ≤ 10%). An unexpected manifestation of MS/MS signal suppression/enhancement was discovered in the course of multi-mycotoxin standard solution stability evaluation.

## 1. Introduction

The degradation of mycotoxins is a desirable process, contributing to the decontamination and safety of food. Food processing, light irradiation, heating, the effect of oxidizing agents and active components of plant origin, fermentation and adsorption are being investigated as measures of mycotoxin control and food safety promotion. Photodegradation is known for aflatoxins (AFLs) [1,2]. The impact of thermal stability and food processing on decontamination was well-reviewed by Milani et al. [3]. Microbial binding and the transformation of mycotoxins is an up-to-date strategy of food and feed detoxification [4]. Contrastingly, for analytical method development and routine analysis stability of the analytes in the stock standard solutions within the storage period and in the calibrants during quantification should be provided.

Meanwhile certified calibrant solutions of mycotoxins are offered for accurate and reliable analysis of a broad spectrum of fungal metabolites, dry standards are much more affordable and for some analytes are the only ones available. They are usually used for method development and validation. Literature data concerning mycotoxins stability in organic solvents and extracts are scattered and devoted to a single or several selected representatives of mycotoxins. The current trend towards the development of multi-analyte methods, however, demands the confirmation of standard solutions stability for dozens of analytes: the whole spectrum of regulated mycotoxins [5,6] or a broad range of fungal metabolites [7,8,9]. Common factors affecting the stability of standard solution are the solvents ability to degrade analyte and to get evaporated, temperature and UV-irradiation. Some analytes can adsorb at the glass vial surface. Moreover, stock standard solutions are periodically brought to room temperature, aliquots are taken and solutions are frozen again. Thus, degradation can occur to mycotoxins during storage and within the execution of analytical procedures. The main objective of the present work was to estimate stability of mycotoxins in the individual stock standard solutions during 14 months of storage at −18 °C and in multi-analyte calibrants under conditions of the chromatographic analysis. The list of analytes under investigation included all regulated mycotoxins, their structural derivatives, some emerging mycotoxins, all in all about three-dozen analytes. The first part of the Results and Discussion section dwells upon the tracking of stability of diluted individual standard solutions in common solvents used in reversed-phase high-performance liquid chromatography (RP-HPLC) and the applicability of UV-spectroscopy to its monitoring. The second part is devoted to the estimation of mycotoxins stability in multi-analyte calibrants under conditions of the routine HPLC analysis.

## 2. Results and Discussion

### 2.1. Stability of Diluted Individual Stock Standard Solutions

UV-spectroscopy is an affordable and straightforward method of tracking mycotoxins stability in individual standard solutions. It is suitable for the quantification of the absolute concentration of mycotoxins exhibiting expressed and selective absorption bands in the spectrum. Otherwise, HPLC coupled with a diode array detector (DAD) or tandem mass spectrometer system (MS/MS) can be used for purity and stability elucidation. The corresponding studies are discussed below, in paragraphs devoted to groups of mycotoxins.

While reviewing available literature, only a few direct pieces of guidance concerning the acceptability of standard solution concentration variation during storage were noted. According to Andrade et al., stock standard solutions of AFLs, ochratoxin A (OTA), zearalenone (ZEA), deoxynivalenol (DON), 3-acetyl- and 15-acetyl deoxynivalenol (3-AcDON and 15-AcDON) and citreoviridin (CTV) were considered valid after a storage period if their UV-spectrophotometric check resulted in a maximum of 3% difference with the previous check of freshly prepared solutions [10]. Some other authors also reported 2%–3% variation as suitable [11,12], while the majority did not address this point. In the present study, the repeatability of intra-day absorbance values was evaluated. It did not exceed 3% for all mycotoxins solutions, but nivalenol (NIV). Thus, variation of absorbance (coefficient of variation, CV) during long-term storage below 3% was considered as solutes stability indicator.

In the present study concentrations of individual stock solutions were selected so that to simplify the preparation of the multi-mycotoxin stock standard solution and HPLC-MS/MS calibrants with respect to mycotoxins maximum (for regulated toxins) or typical contamination levels. To carry out current lab projects vials with individual diluted stock standard solutions were periodically brought to room temperature, unsealed, aliquots were taken, vials were resealed and frozen again. Spectra of all studied individual standard solutions obtained after preparation, in 10 and 14 months of storage can be found in Appendix A. General conclusions about the stability of analytes in standard solutions and samples are presented at the end of each section devoted to the group of mycotoxins. They are based on experimental and literature data. 

#### 2.1.1. Aflatoxins and Sterigmatocystin

AFLs, AFL B1 in particular, are potent hazards, contaminating wide range of crops as secondary metabolites of some *Aspergillus* genera species. They are carcinogenic, and cause growth suppression, immune modulation and malnutrition. High temperatures and humidity favor the accumulation of AFLs. Thus, AFLs are regulated in staple and child food, popular subtropical commodities at 0–40 parts per billion (ppb) levels [13]. Sterigmatocystin (STC) is biogenic precursor of AFL B1 and exhibits similar, though less pronounced, toxic properties. There is no maximum level (ML) set for STC at present, though exposure data are being accumulated [14,15].

The methods for the detection and quantification of AFLs and STC were initially based on normal phase thin layer or high-performance LC; thus, mycotoxins were dissolved in non-polar or intermediate polarity solvents. According to the AOAC Official Methods (1999), AFLs standard solutions are prepared in toluene-acetonitrile mixtures; STC is prepared in benzene [16]. In these solvents, the concentration of AFLs can be confirmed by UV-spectrometry at 350 nm (molar absorptivity ε_AFL B1_ = 19,300, ε_AFL B2_ = 21,000, ε_AFL G1_ = 16,400, ε_AFL G2_ = 18,300 L*mol^−1^*cm^−1^), STC—at 325 nm (ε = 15,200 L*mol^−1^*cm^−1^). Garcia et al. recommended chloroform as a good solvent for the preparation and storage of AFL B1 and G1 [17] and STC [18] standard solutions. Seven-day assay of AFLs solutions (5 μg/mL) at 4 °C revealed 99% and 96% recovery in chloroform, 96% and 83% in methanol [17]. Similar results were obtained for STC in chloroform, while in methanol and acetonitrile, 30 days of cooled or frozen storage resulted in 25%–30% loss [18]. The latter is quite alarming, as methanol is a common solvent for the RP-HPLC applications. 

Beaver found out that approximately two days of storage of AFLs (0.25 μg/mL B1 and G1, 0.075 μg/mL B2 and G2) in methanol-water (50/50 *v/v*) solutions at −18 °C resulted in about a 15% loss of G group AFLs, while the concentration of AFL B1 and B2 declined by a maximum of 6% [19]. Contrastingly, no substantial degradation was observed in 50/50 (*v/v*) acetonitrile-water, acidified acetonitrile- and methanol-water. However, the above solvent mixtures did not provide even 24 h stability of the solutes at room temperature and/or exposed to light. In acidified acetonitrile-water solutions (50/50/0.5 *v/v*), the concentration of mycotoxins was constant for 4.5 h. The positive contribution of acid in solutes stability was possibly due to the decrease in their adsorption on the vial glass surface. Twenty-four-hour monitoring of AFLs degradation in methanol/acetonitrile-water solutions was carried out by Diaz et al. [20]. Pure organic solvent solutions were stable both at 5 and 22 °C. The addition of water induced AFLs degradation: G series degraded more than B, and “1” degraded more than “2”. Only half of the G series AFLs were present in acetonitrile or methanol-water solutions (60/40 *v/v*) after 24 h at room temperature. The authors proposed to overcome the instability of AFLs by keeping them at refrigeration temperature (5 °C) or silanized, or etched with 50% nitric acid glass autosampler vials. The latter probably helped to get rid of the glass surface contaminants that induce AFLs transformation and/or adsorption. 

In the present study, UV-spectra of G series AFLs and STC solutions did not change within 14 months monitoring of methanol standard solutions stored at −18 °C. Increased absorbance in the range of 200–330 nm was observed for B series AFLs. However, within the whole period of observation, the absorbance at λ_max_ of AFLs solutions stayed constant (Table 1). Additional absorption maximum appeared in the spectrum of AFL B2 after ten months of storage. Similar phenomena were noted for DON and OTA standard solutions (address corresponding sections). The variation of absorbance for STC did not exceed 2%. CVs were not calculated for AFLs, because 1 μg/mL AFLs yielded low absorbance at λ_max_. To increase the precision of absorbance measurement, it is desirable to use 5–10 μg/mL standards at least. Nevertheless, absorption spectra in the interval 360 < λ < 380 nm reproduced well within the monitoring period for all studied AFLs, indicating the stability of the mycotoxins.

Thus, experimental data indicate the stability of methanol and acetonitrile standard solutions of AFLs and STC stored at −18 °C for 14 months at least. The literature review suggests that to prevent the degradation of AFLs in water–organic solvent mixtures and the adsorption of analytes on the glass surface, it is recommended to acidify the solution, using of silanized or etched vials to protect samples from light irradiation.

#### 2.1.2. Type A Trichothecenes

Type A trichothecenes belong to the large group of *Fusarium* mycotoxins. They are often detected in cereals, especially oat-containing commodities, and herbal food supplements originating from temperate climate areas [21]. Possessing a similar chemical structure, type A trichothecenes exhibit a similar mode of toxic action. T-2 toxin (T-2) is the most potent, it primarily affects the immune and haematopoietic systems [22]. An ML of 100 μg/kg in food grain was set for this toxin in Russia [23].

Most of the trichothecenes are considered stable, they do not degrade under moderate heating or light irradiation, though chemical transformation in the solution can take place. The available literature suggests that T-2 and HT-2 toxins are stable in organic solvents. Their absorption spectra lack specific bands; thus, HPLC-UV is usually used for both stability and purity estimation. Bennet et al. inspected the purity of T-2 and HT-2 toxins, diacetoxyscirpenol (DAS) and neosolaniol (NeoS) standard solutions using thin layer chromatography (TLC), HPLC-UV, and nuclear magnetic resonance spectroscopy (NMR), and compared their UV spectra in methanol and acetonitrile [24]. The molar absorption of the trichothecenes in acetonitrile was about 2.5 times higher compared to methanol, λ_max_ in acetonitrile solutions was 194-196 nm, and it shifted to 203 nm in methanol. The molar absorptivity of trichothecenes in methanol at 203 nm were ε_T-2_ = 3822, ε_HT-2_ = 1959 (the reference standard contained impurities), ε_DAS_ = 2487, ε_NeoS_ = 2644 L*mol^−1^*cm^−1^. Widestrand et al. monitored these toxins in acetonitrile and ethyl acetate solutions (approx. conc. 10 μg/mL), stored in sealed amber borosilicate glass ampoules at −20 °C, 4 °C, 25 °C and 40 °C for 24 months [11]. Mycotoxins were quantified using reversed-phase HPLC with UV-detection at 195 nm (acetonitrile as an organic component of mobile phase): storage time and the temperature had a negligible influence on toxins concentration. Schothorst declared the two-year stability of acetonitrile standard solutions of nine trichothecenes, including A type T-2, HT-2 toxins and DAS [25]. Flores-Flores monitored the stability of ten trichothecenes in mixed stock standard acetonitrile solution by HPLC-MS/MS: type A trichothecenes T-2, HT-2 toxins, NeoS and DAS (in the range 5–40 μg/mL) were stable for at least 21 days when frozen [26]. Methanol is considered not suitable for the long term storage of trichothecenes at room temperature: their transesterification occurred during storage under such conditions for 22 days [27]. According to Duffy et al., 2 mM (approx. 0.9 mg/mL) of T-2 toxin was stable in 5% aqueous 2-propanol at pH 5.0–6.7 over one year tracked by NMR spectroscopy [28]. The gradual degradation of T-2 toxin into HT-2, T-2 triol and finally T-2 tetraol was detected at a pH of about 11. Similarly, Trusal observed the appearance of breakdown products of ^3^H-labeled T-2, HT-2 toxins, T-2 triol after one and three weeks of storage in Hanks’ balanced salt solution (pH 7.4) at room temperature and 4 °C correspondingly by scanning radioisotopes after thin layer chromatography [29]. 

The Joint Research Centre of the European Commission carried out a collaborative study, aimed at validation of LC-MS/MS method of determination of DON, ZEA, T-2 and HT-2 toxins in cereals and feeds. Prepared sample stability was evaluated. Ethyl acetate extracts and injection solutions (evaporated raw extract reconstituted in water-methanol or water-acetonitrile (1:1, *v/v*)) were stable within a week of storage at 2–10 °C [30]. At least four weeks stability for T-2, HT-2 toxins, NeoS and DAS in frozen evaporated extracts have been reported [26]. 

In the present study T-2, HT-2, NeoS and DAS were dissolved in methanol, T-2 triol in acetonitrile. The studied mycotoxins do not possess any specific chromophores. Thus, we have checked the applicability of UV spectrophotometry to track the constancy of the absorption spectra profiles. Solutions exhibited only one maximum in the range 199–208 nm (Table 2). The spectra profiles of NeoS, DAS and T-2 triol did not change much within the whole period of monitoring, though some increase in absorbance was noted in the region <200 nm. The absorbance of NeoS and T-2 triol individual standard solutions at λ_max_ slightly increased, +0.4% and +6% correspondingly, CVs did not exceed 3%. The absorbance of T-2, HT-2 and DAS standard solutions at λ_max_ increased, for HT-2 up to three-fold (Table 2).

It should be noted that T-2 and HT-2 individual standards were intensely used for the preparation of standard solutions and the fortification of blank samples. Aliquots were taken repeatedly using a pipette equipped with polypropylene tips. Absorbance increase could be due to the contamination of standard solutions by the tips extractives. This conclusion is supported by the increase in absorbance of methanol at 200–240 nm and acetonitrile at 190–230 nm after soaking the pipette tip for 20 min (Appendix A). Thus, it is essential not to use polymeric materials for taking aliquots of standards if UV-spectroscopy is intended to be used for the stability monitoring; chromatographic syringes are a good alternative. Another possible reason for spectra distortion is transesterification of T-2, HT-2 and DAS in methanol.

In conclusion, the experiment and literature review indicate the stability of A-type trichothecenes in acetonitrile for at least two years, even at room temperature. Methanol is best avoided, since transesterification is possible during long-term storage. Acidification can prevent degradation in water–organic mixtures. Prepared samples are better stored in a fridge and seven days shelf life is safe.

#### 2.1.3. Type B Trichothecenes

Type B trichothecenes, as well as A-trichothecenes, are primarily produced by *Fusarium* micromycetes and are common contaminants of grain and grain-based products. Having a common core structure, A- and B-trichothecenes can be differentiated based upon the substitution at the C-8 position. Type B trichothecenes such as DON are generally less cytotoxic than type A, such as T-2 toxin [31]. Human outbreaks from acute exposure to DON lead to symptoms including nausea, vomiting, diarrhea, abdominal pain, headaches, dizziness and fever; however, the evidence of adverse health effects in humans due to chronic exposure to DON and its analogues is lacking [32]. MLs set for DON in EU are from 200 to 750 μg/kg for different kinds of cereal products [33].

In contrast to the type A-trichothecenes, the B-trichothecenes have a conjugated carbonyl group at C-8 generating absorption band of UV spectra. According to the AOAC Official Method procedure (1984), the concentration of DON standard solutions in methanol can be estimated by UV-spectroscopy (ɛ = 6384 L*mol^−1^*cm^−1^) [34]. Similar to A type trichothecenes, poor specificity at the low wavelength increases the risks of interferences. HPLC-UV is preferable instrument for quantification. Shepherd et al. monitored DON standard solutions stability at 220 nm [35]; Visconti et al. detected NIV, DON, fusarenone X (FusX) and 3-AcDON at 225 nm [36]; Kotal and Ok—at 218 nm [37,38]; Yang—at 224 nm [39]. Within the European Commission project aimed at producing and certifying calibrants for the determination of B-trichothecenes, Krska et al. applied a wide spectrum of analytical techniques for the estimation of purity and contaminants levels in standard solutions of DON, 3-AcDON, 15-AcDON and NIV [40]. Absorption spectra yielded λ_max_ values of 216 nm, 217 nm, 219 nm and 217 nm, respectively. However, the authors consider purity estimations based on general molar absorptivity (ɛ = 6400 L*mol^−1^*cm^−1^) can yield uncertainty caused by differing molar absorptivity of impurities and mycotoxins. Contrastingly, Vidal et al. followed the AOAC Official methods of analysis to check DON concentration in standard solutions by UV-spectroscopy [41,42]. 

Shepherd estimated the long-term stability of DON standard solutions in ethyl acetate, acetonitrile and chloroform/methanol (95/5 *v/v*) (concentration range 25–100 μg/mL), stored in fire-sealed glass ampoules at −36 °C, 4 °C, ambient temperature and +37 °C in the dark for 27 months. Ethyl acetate proved to be the best solvent for the long-term storage of DON standard solutions even at an elevated temperature. Chloroform/methanol proved to be the least suitable. Appropriate for RP HPLC acetonitrile caused progressive decomposition of the toxin when solutions were stored in a fridge, at ambient or elevated temperature. Freezing provided a constancy of DON concentration in acetonitrile solutions for a minimum of 27 months [35]. Widestrandt’s results were the opposite: temperature (−18 °C to +40 °C) and storage time (up to 24 months) did not affect DON and NIV (approx. 10 μg/mL) in acetonitrile solutions. The gradual degradation of mycotoxins in ethyl acetate solution took place, especially at elevated temperatures. The decreases in DON and NIV concentrations after 24 months of storage in ethyl acetate at 4 °C were 21% and 11% correspondingly [11]. Schothorst et al. declared mixed acetonitrile standard solutions of DON, 3- and 15-AcDON, NIV and FusX (conc. from approx. 0.4 to 3.0 μg/mL) to be stable for at least two years when refrigerated (+4 °C) [25]. Jensen prepared stock solutions of DON, 3- and 15-AcDON in acetonitrile also [43]. Stock and mixed standard solution, containing 10–40 ng/mL analytes, were stored at −18 °C. The latter was renewed monthly. 

NIV and DON proved to be relatively stable in the aqueous buffer solutions over the pH range of 1–10, at pH 10 there was <20% degradation after 26-day exposure under ambient conditions. A higher decomposition rate was observed at elevated temperature (80 °C) [44]. No DON transformation was expected to occur under standard storage or analysis conditions (moderate temperature (<120 °C), acidic or neutral pH, absence of oxidizing reagents) [45]. Breidbach et al. tested the stability of DON in raw cereal extracts and injection solutions in acetonitrile–water–formic acid (80/19.9/0.1 *v/v* and 50/9.9/0.1 *v/v* correspondingly) and estimated that they were stable at 2–10 °C for at least seven days in deactivated vials [30]. Bucheli et al. detected DON in river water samples, amd the evaporated extract was reconstituted in water–MeOH (90/10 *v/v*), stored in amber glass vials at 4 °C and analyzed within 48 h [46]. 

In the present study, stability of B-trichothecenes in methanol and acetonitrile individual standard solutions was tested. Absorption spectra yielded λ_max_ at 217 nm (NIV), 218 nm (DON), 208 nm (3-AcDON), 221 nm and 219 nm (15-AcDON) and 216 nm and 218 nm (NIV) (Table 3). An absorption maximum at 284 nm indicating the presence of contaminants was observed for DON in methanol. UV spectra of DON and NIV in methanol were almost constant, while acetonitrile solutions yielded poor spectra, that changed much during standard solution storage. UV spectra of 3-, 15-AcDON and FusX in methanol and acetonitrile were similarly shaped and stayed constant within 14 months storage. Absorption at λ_max_ for almost all B-trichothecenes increased in 2–8%. 3-AcDON in acetonitrile was the exception. Its absorption has fallen by 10%. Spectra of DON and NIV acetonitrile solutions changed dramatically. The latter can be due to mycotoxin degradation or accidental standard contamination. 

According to available literature data, B trichothecenes are stable in acetonitrile at −18 °C for two years at least. On the contrary, the repeatability of DON and NIV spectra in acetonitrile solutions was poor in our experiment. CVs of the absorbance at λ_max_ for 3-AcDON in acetonitrile, DON, 15-AcDON in methanol were over 3%. Spectra profiles stayed constant. Thus, these solutions could be used for the purpose of method development, but the concentration of mycotoxins should be traced and corrected. In general, methanol solutions of type B trichothecenes proved to be more stable and suitable for monitoring by UV-spectrophotometry. Literature data suggest calibrants and extracts in water–methanol/acetonitrile mixtures should be prepared in deactivated vials, acidified and stored in the fridge. Seven days storage was demonstrated to be safe.

#### 2.1.4. Fumonisins

Fumonisins (FBs) are predominantly secondary metabolites of *Fusarium* spp. Their highest amounts are found in maize and maize-based products. Health issues linked to FBs are cancer of the esophagus, hepatocarcinoma, the stimulation and suppression of the immune system, and defects in the neural tube [47]. MLs set for the sum of FBs set in EU are from 200 to 1000 μg/kg for different kinds of maize-based products [33].

FBs lack chromophores providing their UV detection. Thus, this section is based on literature data only. Traditionally, FBs are monitored by HPLC after their pre-column derivatization with o-phthalic aldehyde [48,49]. This method was used by Visconti et al. to monitor methanol and acetonitrile-water (50:50, *v/v*) FB1 (100 μg/mL) and FB2 (50 μg/mL) standard solutions during storage in fire sealed glass ampoules for six months [12]. Methanol solutions proved to be stable under freezing for six weeks storage, at 4 °C their degradation started in three weeks, about 25% loss was detected within this period at ambient temperature. The authors recommended acetonitrile-water (1/1, *v/v*) as a good solvent for long-term FBs storage, even at ambient temperature. Chilaka et al. utilized mixed methanol standard solution, containing FB1, FB2 and FB3 while investigating the effect of food processing on mycotoxins stability. The mixture was stored at −18 °C [50].

The heating of aqueous buffered FB1 solution (5 mg/mL, pH 4) for 80 min at 100 °C did not result in mycotoxin transformation [51]. The heat stability of FBs at the temperature below 100–120 °C is supported by the literature reviewed by Humpf [52].

Thus, FBs can be stored in acetonitrile/water (50/50 *v/v*) even at ambient temperature. Methanol standard solutions should be frozen. Aqueous solutions are stable.

#### 2.1.5. Zearalenone and Its Derivatives

ZEA contaminates cereal crops worldwide, particularly in temperate and warm regions. Being *Fusarium* metabolite, it usually co-occures trichothecenes. It has expressed oestrogenic effects in mammalians. The biotransformation of ZEA results in α-, β-zearalenol (α-, β-ZEL) and α-, β-zearalanol (zeranol, taleranol; α-, β-ZAL). Interestingly, α-ZEL exhibits greater estrogenic activity relative to the parent compound. ZEA derivatives are produced by fungi also, but at much lower levels [53,54]. MLs set for ZEA in cereals and processed cereal-based foods in the EU vary from 20 to 200 μg/kg [33].

ZEA and its derivatives possess chromophores yielding specific absorption bands. The UV-spectra of ZEA, α- and β-ZEL methanol solutions show similar absorption maxima (at about 236, 274 and 316 nm). The first UV maximum is shifted to 238 nm for β-ZEL [24]. According to Nesheim et al., molar absorbtivity of ZEA in methanol at 314 nm is equal to 6400 L*mol^−1^*cm^−1^ [16]. Absorption at 236 nm was utilized by Bennet et al. to track the purity and stability of ZEA and its metabolites by HPLC-UV [24]. The authors reported the storage of standard methanol solutions of ZEA, α- and β-ZEL (10 μg/mL) to be successful within 18 months when protected from light and refrigerated (4 °C). Heat stability of ZEA in aqueous buffer solutions was studied by Ryu et al. by HPLC using fluorimetric detection. Regardless of pH, no significant losses in ZEA occurred during processing at 100 °C within about 80 min [55].

The present study demonstrated that ZEA acetonitrile solution was stable under storage conditions: absorbance variation at λ_max_ = 273 nm did not exceed 1.4%. Concentrations of α-, β-ZEL and α-, β-ZAL standard solutions decreased: the estimated decline depended on the absorption maximum wavelength. Average loss was 11% and 19% for α-ZEL, 11% and 34% for β-ZEL, 20% and 22% for α-ZAL, 19% and 21% for β-ZAL for 10 and 14 months storage correspondingly (absorption at 218 nm was not taken into account for the last two mycotoxins) (Table 4).

Thus, following the literature data, the present study confirmed ZEA acetonitrile standard solution stability within 14 months storage at −18 °C. Gradual degradation was observed for ZEA derivatives in acetonitrile. Meanwhile, according to literature data, methanol standard solutions of α- and β-ZEL do not degrade at −4 °C for at least 18 months. 

#### 2.1.6. Ochratoxin A

Certain species of *Aspergillus* and *Penicillium* produce OTA. It is found in a variety of foods including grains and grain products, preserved meats, fresh and dried fruits, nuts, coffee, wine. OTA is also found in breast milk. It has been shown to be nephrotoxic, hepatotoxic, teratogenic and immunotoxic. OTA accumulates in the kidney and is particularly toxic to this organ [56]. The MLs set for OTA in different kinds of food in the EU vary from 0.5 to 10 μg/kg [33].

According to the AOAC Official Methods (1995), OTA standard solution is prepared in benzene-acetic acid (99/1 *v/v*), and its concentration is confirmed by UV-spectrometry at 333 nm [16]. At present, RP HPLC is a method of choice for OTA determination in complex matrixes; nevertheless, many analysts prefer to store high concentration stock standard solutions in acidified non-polar solvents (benzene, toluene) matched by UV-spectrometry (ε_333_ = 5440 [toluene-acetic acid] and ε_333_ = 5600 [benzene-acetic acid] L*mol^−1^*cm^−1^). To prepare calibrant solutions for RP-HPLC, the aliquot of stock standard solution is evaporated to dryness under stream of nitrogen at room temperature and redissolved in RP-HPLC suitable solvent [57,58,59,60]. 

Spectral characteristics of OTA in RP HPLC solvents are also available. Lippolis et al. used UV-spectrometry to specify the concentration of 10 μg/mL methanol OTA standard solution (ε_332_ = 6330 L*mol^−1^*cm^−1^) [61]. Igarashi et al. tested the stability of methanol (C_OTA_ = ~0.01–10 μg/mL) and methanol-water-acetic acid (30/70/1, *v/v*) (C_OTA_ = 2 and 10 ng/mL) standard solutions by HPLC-FLD or –MS. They were stable when kept at about 5 °C, OTA concentration was constant for at least 50 days storage [62].

Liazid et al. studied OTA stability in methanol standard solution (~50 μg/mL) subjected to various manipulations used for sample preparation. OTA concentration was tracked by HPLC-FLD. Stirring of solution at 25–60 °C for 20 min with or without ultrasound assistance and microwave assisted extraction at temperatures up to 150 °C for 20 min have not led to any significant OTA concentration change. The maximum working temperature of pressurized liquid extraction appeared to be 100 °C for the same time interval [63]. 

In the present study, 1 μg/mL OTA methanol solution was used as a standard for multi-mycotoxin calibrant preparation. This concentration proved to be too low for stability elucidation: absorbance at 333 nm (λ_ref_ = 400 nm) comprised several hundredths of absorbance unit (0.02–0.03 within the whole storage period), moreover, storage resulted in the appearance of an additional maximum at approx. 290 nm, that overlapped absorption at 333 nm (Appendix A). It is advisable to use at least 10–25 μg/mL OTA solutions for stability tracking.

Thus, according to the available literature, OTA is stable in methanol for at least 50 days when stored at about 5 °C. 

#### 2.1.7. *Alternaria* Toxins

Alternariol (AOH), its monomethyl ether (AME), altenuene (ALT) and tentoxin (TE) are representatives of a vast group of phytotoxins produced by *Alternaria* spp. These fungi are common contaminants of cereal crops, including sunflower seeds. They are responsible for the spoilage of harvested fruits and vegetables even during refrigerated transport and storage. Few *Alternaria* toxins have been characterized. In vivo studies are limited. There are data on the cytotoxicity of *Alternaria* toxins, and their immunomodulating and estrogenic properties. Their mean concentration in contaminated grain can reach dozens to thousands ppb [64,65]. These mycotoxins are not regulated in food.

Zwickel et al. ascertained the stability (±10%) of methanol stock solutions (1–0.5 mg/mL) and of *Alternaria* toxins when stored at −30 °C over two years. The actual concentration of mycotoxins was determined by UV-spectrometry: AOH and AME in acetonitrile at λ_max_ = 256 nm (ε = 40,600 and 47,600 L*mol^−1^*cm^−1^), ALT in ethanol at λ_max_ = 240 nm (ε = 30,000 L*mol^−1^*cm^−1^) and TE in 95% ethanol at λ_max_ = 282 nm (ε = 20,700 L*mol^−1^*cm^−1^) [66]. Molar extinction coefficients for AOH and AME in acetonitrile used by Zwickel were earlier calculated by Asam et al. [67]. In the latter paper, the authors estimated the stability of deuterated AOH and AME in acetonitrile/water and acetonitrile/buffer (pH 2–9). Beyond pH 9 decomposition of AOH occurred after three weeks of storage. At a lower pH, standard solutions proved to be stable at room temperature for over three weeks, this might also be true for non-deuterated AOH and AME.

The present study demonstrated that AOH, AME, ALT and TE are stable in methanol (acetonitrile) solutions under storage conditions. Absorbance variation at all λ_max_ did not exceed 3% (Table 5). This is in accordance with literature data indicating the high stability of frozen methanol and acetonitrile standard solutions of AOH and AME. They can be stored for at least two years. In aqueous organic solutions, stability lasting more than three weeks is maintained by acidification.

#### 2.1.8. Enniatins A and B, Beauvericin

Enniatins (Enns) and beauvericin (BEA) are produced mainly by *Fusarium* fungi under moist and cool conditions. They are often detected in grain, dry fruits, chicken meat, and herbs. The antimicrobial and antibiotic properties of these compounds are known; they can act as enzyme inhibitors, and as compounds inducing oxidative stress. They also have cytotoxic activity towards different cell types, inducing apoptosis. Their contamination levels in grain and herbs are usually in μg/kg, sometime reaching mg/kg range [65,68]. Enns and BEA are not regulated in any food.

BEA and Enns are only short wavelength-absorbing: BEA can be quantified by HPLC-UV at 192–225 nm, Enns are usually detected after derivatization or by MS/MS [69,70]. The latter was used by Taevernier et al. to trace Enns and BEA stability in acetonitrile (ethanol)-water solutions (10%, 50% and 95% *v/v* organics). They were stored in glass vials protected from light at −35 °C, 5 °C, 25 °C and 40 °C for seven days. No significant degradation was observed even in the worst-case stability scenario [71].

Absorption spectra of Enns and BEA in individual methanol solutions obtained in the present study were similar; the only maximum was observed at 206–207 nm, band width 190–250 nm. Storage did not affect the spectrum shape or absorbance values (Table 6). 

Thus, Enns and BEA can be considered stable in methanol when frozen for at least 14 months. Literature data suggest that these mycotoxins do not transform in diluted calibrants and sample extracts for at least seven days, even at the room temperature.

#### 2.1.9. Moniliformin, Mycophenolic Acid, Citrinin, Citreoveridin, Cyclopiazonic Acid

Moniliformin (MO) is mainly produced by *Fusarium* spp. It contaminates grain and grain products at the level of dozens to hundreds ppb, sometimes up to part per million (ppm). MO is acutely toxic; the heart is the main target organ. It causes heart failure, muscle weakness, respiratory distress, and negatively affects immunity and animal performance. MO is not regulated in any food [65,72]. Citrinin (CIT) can be produced by many fungal species belonging to the genera of *Penicillium, Aspergillus* and *Monascus.* It is detected in cereals, red yeast rice, fruits, herbs, and spices in the range of dozens to hundreds of ppb. It is nephrotoxic [73]. An ML was set for CIT in food supplements based on rice fermented with red yeast *Monascus purpureus* at 100 μg/kg [74]. Mycophenolic acid (MPA) is mainly produced by *Penicillium* spp. Its acute toxicity is low. At present, it is widely used both as an immunosuppressive drug for prophylaxis and the treatment of organ rejection in transplantations and as an antirheumatic drug. Unwanted immunosuppression is a major concern. It can be found in various foods in higher concentrations compared to other fungal metabolites, reaching ppm levels. MPA is not regulated in food [75]. Available information on CTV and cyclopiazonic acid (CPA) is scarce. CTV is produced by *Pennicilium citreonigrum* on rice, with contamination levels in dozens of ppb. It is connected with cases of beriberi [76]. The main CPA producers are *Aspergillus* and *Penicillium* spp. It exhibits cytotoxicity and immunotoxicity on human cells. It was detected in grain, vegetables, and cheese at ppb levels [77].

According to Nielsen, all these mycotoxins have pronounced characteristic spectra with maximums at 228 nm and 260 nm for MO; 216 nm, 252 nm and 304 nm for MPA; 216 nm and 328 nm for CIT; 204 nm, 236 nm, 296 nm and 388 nm for CTV; 224 nm and 280 nm for CPA [70]. According to the AOAC Official Methods (1999) the CIT standard solution is prepared in chloroform, CPA is dissolved in methanol and water is the best solvent for MO [13]. For CIT ε_332_ = 16,100 L*mol^−1^*cm^−1^, for CPA ε_284_ = 20,417 L*mol^−1^*cm^−1^, for MO ε_260_ = 5600 L*mol^−1^*cm^−1^ in the corresponding solvents. Reinhard reported absorption maximum of CIT in methanol at 321 nm, ε = 20,700 L*mol^−1^*cm^−1^ [78]. Data concerning stability of these mycotoxins in the standard solutions under storage or analysis conditions are scarce. Filek et al. reported water solution of MO (1 mg/mL) to be stable for at least three months when refrigerated [79]. Furthermore, 1–75 ng/mL MO calibrants dissolved in aqueous 0.1% acetic acid underwent about 50% degradation within two weeks storage at 4 °C and the authors recommended practicing the fresh preparation of MO calibrants as a quality control strategy [80]. CIT (0.5–100 μg/mL) methanol standard solutions were stable at room temperature for 24 h [81]. The transformation of CIT under heating in acetonitrile, water and methanol solutions was found out by Xu et al. [82]. Standard solution (2 μg/mL) vaporization at room temperature yielded about 50% degradation ratio. The authors suggested determination procedures to be carried out under low temperatures (<4 °C). 

In the present study, CPA, CTV and MPA spectra were recorded immediately after preparation of individual standard solutions and in four months of storage, the CIT and MO spectra were recorded after preparation, following 10 and 14 months of storage, respectively. Absorbance variation at selected λ_max_ did not exceed 2% for all four analytes, but MO (Table 7). CV for MO was 6.4%, its spectrum profile changed much only in the low UV region below 210 nm. 

Thus, CPA, CTV and MPA methanol individual standard solutions proved able to preserve stability when stored at −18 °C for at least four months; whilst CIT preserved stability for 14 months. MO standard solution stability should be monitored, and mycotoxin concentration corrected if necessary. It should be pointed out, that according to literature data, MO tends to degrade in calibrants and samples, the analysis should be carried out as quickly as possible. Analyses of CIT transformants should take place in solution under heating or solvent vaporization.

### 2.2. Multi-mycotoxin Diluted Standard Solutions Stability

Monbaliu et al. stored methanol stock standard mixture of mycotoxins (A- and B-trichothecenes, Alternaria toxins, FBs, AFLs and STC, ZAN, ZEN, BEA; 0.1–7.5 μg/mL) at −18 °C for three months [83]. Slobodchikova et al. proposed to store combined multi-mycotoxin (A- and B-trichothecenes, FBs, AFLs, ZEA, α- and β- ZAL and ZEL, 10 μg/mL each) stock methanol solution in aliquots at −80 °C for six months maximum [84]. Calibration solutions are usually freshly prepared from a stock standard mixture [85]. They are usually diluted down to ppb and part per trillion levels in water-organics. Sequence execution may take several hours and even days; it is therefore useful to evaluate mycotoxins stability in calibrants under analysis conditions. 

The stability of analytes in diluted multi-mycotoxin standard solutions was studied for AFLs, STC, A- and B-trichothecenes, ZEA, α- and β- ZEL, OTA, Alternaria toxins, Enns and BEA, CIT and CTV. Guided by the literature recommendations in Section 2.1, calibrants were diluted with methanol/water/formic acid (50/50/0.1 *v/v*). To prevent the adsorption of analytes on the glass surface of chromatographic vials, diluted calibrants were dispensed in silanized inserts. The worst scenario was used for stability evaluation: autosampler temperature was set to 23 °C, light “on”. Three series of 7–10 injections were executed. Diluted stock multi-mycotoxin standard solution was injected every two hours within one sequence. All in all, the experiment lasted 75 h. The control calibrants, that have been stored at −18 °C, were brought to room temperature and injected in triplicate. The average areas of chromatographic peaks at the end of stability experiment and controls coincided within peak area variation, calculated as average CV for three injections series (Figure 1, Appendix A). CVs of MS/MS signal did not exceed 10% for all studied mycotoxins, except CIT. Its concentration in 10-fold diluted mixed standard solution exceeded the linearity range resulting in low repeatability. Thus, no dramatic (>10%) degradation of mycotoxins could be noted within 75 h HPLC-MS/MS monitoring. The repeatability of analytical signal of mycotoxins in 100-fold diluted mixed standard solution was poorer: for seven analytes (NIV, DON, FusX, T-2 triol, AFL G1, AFL B2, Enn B) it exceeded the 10% margin. Within CV, the coincidence of tests and controls cannot be used for the estimation of the stability of these analytes in diluted standard solution.

It is interesting that for the majority of analytes, the chromatographic peak areas changed gradually within 75 h of stability monitoring. The relative increase in analytical signal was noted for strongly retained analytes; early eluting mycotoxins were subjected to increasing suppression (Figure 2, corresponding data is available in Appendix A (Figure 2 full page size), Appendix A). 

For example, the analytical signals of NIV, 15-AcDON and 3-AcDON in 100-fold diluted multi-mycotoxin standard solution decreased by ≥30%. Only for NIV was the decline within signal variation (CV, %). On the other hand, the analytical signals of OTA, ZEA, EnnA increased by over 10%, and the tendency exceeded signal variation. MS/MS signal suppression/enhancement (SSE) within sequence execution can result in calibration curve splitting—that is, a mismatch of calibration curves obtained at the beginning and at the end of sequence [86]. Usually, SSE in HPLC-MS/MS accounts for the presence of co-eluting compounds in the sample matrix. Matrix-matched calibration, internal standards, sample dilution, and selective sample preparation procedures are utilized to overcome this unwanted phenomenon [87,88]. In this study, negative/positive trends in signal variation were observed without any matrix, as analytes were dissolved in neat solvents. Thus, a kind of matrix effect was observed in the absence of sample matrix.

## 3. Materials and Methods

### 3.1. Standards

Individual neat analytical standards of AFL B1 (purity ≥ 98%), AFL B2, AFL G1, AFL G2, STC (≥98%), T-2 toxin (≥98%), HT-2 toxin (≥98%), DAS, NIV (98,2%), DON, (≥98%), 3-AcDON, 15-AcDON, CIT (≥98%), FusX (98,7%), FB1 (≥98%), FB2 (≥98%), ZEA, (≥98%), α- and β-ZEL, α- and β-ZAL, OTA (≥98%) were purchased from Sigma Aldrich (Moscow, Russia); AOH (99,3%), AME (99,77%), ALT (98%), BEA (99,31%), CTV (97%), CPA (98%), EnnA (99,68%), EnnB (99,62%), MPA (99,59%), MO (99%), NeoS (99%), T-2 triol (99%), TE (99,84%) were supplied by Fermentek (Jerusalem, Israel). Stock solutions were prepared in acetonitrile (AFLs, STC, A and B trichothecenes except NeoS, CIT, ZEA and analogues, OTA, MPA), methanol (*Alternaria* toxins, Enns and BEA, CTV, CPA, MPA, MO, NeoS) or acetonitrile/water, 50/50, *v/v* (FB1, FB2) at a concentration of 100 or 500 μg/mL by exact weighing of mycotoxins obtained in powder and dissolved in 10 mL of solvent. All stock solutions were stored in amber glass vials sealed with parafilm at −18 °C. 

The selection of solvent and concentration of diluted individual stock solutions intended for the preparation of multi-mycotoxin standard was based on a review of the literature concerning physical properties and occurrence of mycotoxins. Solvents used in the present study, and the concentration of diluted individual stock solutions for all mycotoxins except FB1 and FB2 are presented in Table 1, Table 2, Table 3, Table 4, Table 5, Table 6 and Table 7 of Results and Discussion (Section 2.1). All individual diluted stock solutions were stored in amber glass vials sealed with parafilm at −18 °C. FB1 and FB2 diluted individual stock solutions were prepared in acetonitrile/water, 50/50, *v/v* (FB1, FB2) at a concentration of 10 mg/mL and stored at −4 °C. Before use, the standard solutions were brought to room temperature and stirred. These standard solutions were utilized for preparation of various standard solutions of mycotoxins within projects carried out in the laboratory as well as stability monitoring.

The multi-mycotoxin stock standard solutions of AFLs B1 and G1 (0.1 μg/mL each), AFLs B2 and G2 (0.025 μg/mL each), STC (0.1 μg/mL), T-2 (0.8 μg/mL), HT-2 (8 μg/mL), T-2 triol (5 μg/mL), NeoS, DAS, NIV, DON, 3-AcDON, 15-AcDON, FusX (1 μg/mL each), FB1 (2 μg/mL), FB2 (0.6 μg/mL), ZEA (2 μg/mL), α- and β-ZEL (5 μg/mL each), OTA (0,1 μg/mL), AOH, AME, ALT (2 μg/mL each), TE (0.8 μg/mL), EnnA, EnnB and BEA (0.2 μg/mL each), MO (1 μg/mL), MPA (10 μg/mL), CIT (1 μg/mL), and CTV (5 μg/mL) were prepared in methanol and stored in amber glass vials sealed with parafilm at −18 °C. 

Multi-mycotoxin stock standard solution was diluted 10- and 100-fold with methanol-water-formic acid (50/50/1 *v/v*) and immediately used for the monitoring of diluted multi-mycotoxin standard solutions stability. 

### 3.2. Reagents and Materials

Water was purified using a Milli-Q system (Millipore, USA). Formic acid (pure, 98+%, Acros organics, Germany), HPLC-grade methanol (LiChrosolv, Merck, Germany) and acetonitrile (Panreac, AppliChem, Germany) were used for the preparation of standard solutions of mycotoxins and mobile phases. Potassium dichromate solution supplied by Sigma-Aldrich (Moscow, Russia) was used as the UV-VIS standard for spectrophotometer calibration.

### 3.3. Stability Experiments

#### 3.3.1. UV Spectrophotometry of Individual Standards

UV spectra of individual diluted stock standard solutions of mycotoxins were recorded immediately after their preparation, at 10 and 14 months of storage and utilization. An Agilent HP 8453 spectrophotometer (± 1.0 nm, ±1.0% T) was calibrated before each series of measurements using potassium dichromate solution. Individual mycotoxin standard solution absorption spectrum was taken in triplicate minimum, absorbance at λ_max_ presented in the Table 1, Table 2, Table 3, Table 4, Table 5, Table 6 and Table 7 is an average result of parallel measurements. Intra-day variance was calculated for each standard solution. It did not exceed 3% for all studied mycotoxins in solutions besides NIV (4.8%). Thus, mycotoxin concentration was considered constant if variation of average absorbance at λ_max_ within the storage period did not exceed 3%.

#### 3.3.2. HPLC-MS/MS of Multi-Mycotoxin Standard Solution

The HPLC analysis was performed with a Vanquish UHPLC system consisting of a binary pump, autosampler and column compartment combined with a triple quadrupole mass spectrometer with a heated electrospray source TSQ Endura controlled by Xcalibur 4.0 QF2 Software (all Thermo Scientific, USA). HPLC column Ascentis Express F5 (100 × 3.0 mm I.D. pore size 90 Å, particle size 2.7 μm) was supplied by Supelco (Sigma Aldrich, Moscow, Russia). Self-made post-column T-splitter delivered 10% effluent to detector, split ratio 1:9. Column compartment temperature was 25 °C. The autosampler chamber temperature was set to 23 °C, light “on”.

The mobile phase composition and gradient conditions were optimized to achieve a suitable resolution of chromatographic peaks. It was composed of water (solvent A) and methanol (solvent B), both containing 0.1% (*v/v*) formic acid. A gradient program was set up as follows: 0–15 min linear gradient from 40 to 95 % B; hold at 95% B for 7 min; return to 40% B in 1 min and hold at 40% B for 4 min (total run time 27 min). 

MS/MS detection was performed in positive electrospray ionization mode. Flow injection analysis set up was used for the optimization of MS/MS detection of individual mycotoxins, scheduled MRMs are provided in Appendix A. Optimized source parameters: vaporizer temperature 225 °C, positive/negative spray voltage ±4500 V, ion transfer tube temperature 200 °C, sheath gas – 35, aux gas – 10, sweep gas – 2 arb. units; CID gas (argon) 2 mTorr; dwell time 100 ms, Q1 and Q3 resolution 0.7 FWHM. 

Multi-mycotoxin stock standard solution was diluted 10- and 100-fold by a mixture of A and B HPLC phases (50/50, *v/v*). Diluted multi-mycotoxin standard solutions were dispensed in chromatographic vials with inserts (400 mL silanized flat bottom inserts supplied by Agilent (CA, USA)), so as to gain the following sets of standard solutions: (A) 10- and 100-fold diluted multi-mycotoxin stock solutions for stability testing and (B) the same as (A), but kept frozen until the end of stability experiment and used for testifying the constancy of the MS/MS detection. Three days of stability monitoring was carried out for A set standard solutions: 7 consequent injections with about 1.5 h intervals, 10 h in standby mode; 10 consequent injections with 2 h intervals, about 20 h in standby mode; 10 consequent injections with 2 h intervals. B set calibrants were injected immediately after the stability experiment. 

Intra-day analytical signal variance (CV, %) was calculated for each series of injections; an average of 3 series was considered as characteristic for an analyte. The average areas of chromatographic peaks obtained for the last 5 injections of A set standard solutions at the end of stability monitoring experiment and 3 injections of B set calibrants were compared. 

## 4. Conclusions 

Long-term storage and monitoring of mycotoxins in individual standard solutions in RP-HPLC solvents by UV-spectroscopy revealed several groups of analytes:

1. The spectrum exhibits characteristic band(s), its profile and absorption at maximum (a) are constant (CV<3%). These solutions can be referred to as stable under storage conditions: aflatoxins G1 and G2, sterigmatocystin, alternariol, altenuene, tentoxin, citrinin, citreoveridin, cyclopiazonic acid in methanol and alternariol methyl ether, zearalenone, mycophenolic acid in acetonitrile. 

2. The spectrum exhibits characteristic band(s), its profile does not change much, absorption at maximum(a) changes (CV>3%). Solvent evaporation or gradual degradation occur. Concentration can be adjusted basing on UV-spectroscopy monitoring: moniliformin in methanol; zearalenone derivatives group: α-, β-zeralanol and α-, β-zearalenol.

3. The spectrum exhibits characteristic band(s), its profile changes, absorption at maximum(a) are constant (CV<3%). New absorption bands appear. Contamination is a possible reason. Hence characteristic band pattern and absorption values are constant; the solutes are referred to as stable. Literature data confirms stability: aflatoxins B1 and B2, ochratoxin A.

4. Absorption band is not selective, spectrum profile and absorption at maximum are constant (CV<3%). These solutions can be referred to as stable under storage conditions: methanol solutions of enniatins A and B, beauvericin, neosolaniol, nivalenol, 3- and 15-acetyldeoxinivalenol, fusarenone X; 15-acetyldeoxinivalenol, fuzarenone X and T-2 triol in acetonitrile.

5. The absorption band is not selective, spectrum profile change. UV-spectroscopy is not applicable for stability monitoring. Degradation or contamination occurs: deoxynivalenol, T-2, HT-2 toxins and diacetoxyscirpenol in methanol; nivalenol, deoxynivalenol and 3-acetyldeoxinivalenol in acetonitrile. Literature review suggests acetonitrile is preferable for long-term storage of trichothecenes A and B.

The impact of solvent evaporation and the contamination of solutions with low-UV absorbing extractives from polypropylene pipette tips used for taking aliquots was noted for all frequently used individual stock standard solutions. To prevent these, it is preferable to store aliquots of standard solutions in flame-sealed glass ampoules or use chromatographic syringes to take aliquots of stock standards.

10-fold diluted multi-mycotoxin standard in water/methanol (50/50 *v/v*) solutions acidified with 0.1% formic acid proved to be stable within 75 h in silanized glass at 23 °C exposed to light (CV ≤ 10%). The repeatability of the HPLC-MS/MS analytical signal of some analytes in 100-fold diluted multi-mycotoxin standard solution was low. Stability of this calibrant could not be evaluated. An unexpected manifestation of MS/MS signal suppression/enhancement was discovered in the course of multi-mycotoxin calibrant stability evaluation.

## Figures and Tables

**Figure 1 toxins-12-00094-f001:**
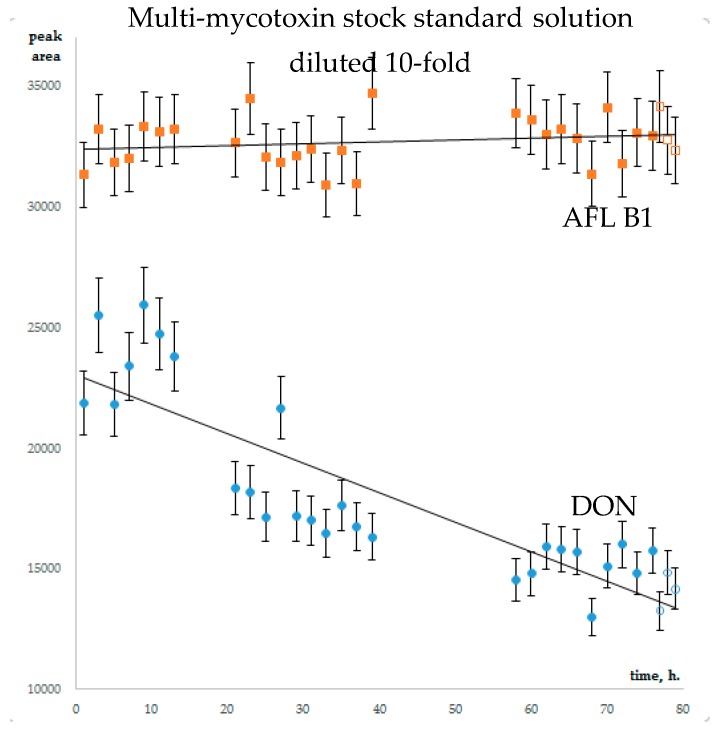
AFL B1 and DON chromatographic peaks area change within the stability experiment. Full symbols – test, open – control.

**Figure 2 toxins-12-00094-f002:**
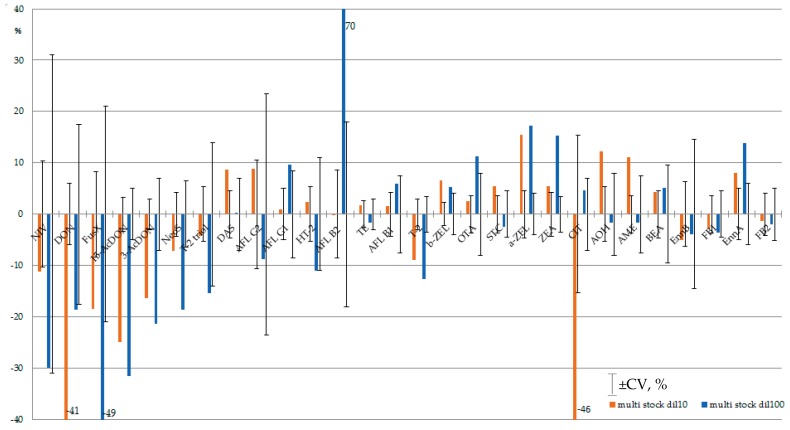
Suppression/enhancement of MS/MS analytical signal within 75 h of stability monitoring of multi-mycotoxin stock standard solution diluted 10- and 100-fold (brown and blue correspondingly). Error bars indicate average coefficient of variation of analytical signal of mycotoxin in 10- and 100-fold diluted multi-analyte standard solution (CV, %). Mycotoxins are presented in their elution order.

**Table 1 toxins-12-00094-t001:** The absorbance of AFLs and STC individual standard solutions.

Mycotoxin	Solvent	Conc., μg/mL	λ_max_/λ_ref._, nm	A (Average, *n* = 3)		
				Fresh	10 Months	14 Months
AFL B1	MeOH	1	360/400	0.07	0.07	0.07
AFL B2	MeOH	1	360/400	0.08	0.08	0.08
AFL G1	MeOH	1	365/280	0.05	0.05	0.05
AFL G2	MeOH	1	365/280	0.05	0.05	0.05
STC	MeOH	21.5	325/280	0.86	0.83	0.85

MeOH—methanol, AFL—aflatoxin, STC—sterigmatocystin.

**Table 2 toxins-12-00094-t002:** The absorbance of type A trichothecenes individual standard solutions.

Mycotoxin	Solvent	Conc., μg/mL	λ_max_/λ_ref._, nm	A (Average, *n* = 3)			CV, %
				Fresh	10 Months	14 Months	
T-2	MeOH	50	203/250	0.72	0.99 (+38%)	1.01 (+40%)	17.9
HT-2	MeOH	20	202/220	0.32	0.49 (+53%)	1.20 (+275%)	69.7
T-2 triol	ACN	200	199/250	2.85	2.74 (−4%)	2.86 (+0,4%)	2.4
NeoS	MeOH	200	208/250	1.94	2.00 (+3%)	2.06 (+6%)	3.0
DAS	MeOH	50	203/300	0.93	1.05 (+13%)	1.07 (+15%)	7.5

MeOH—methanol, ACN—acetonitrile, NeoS—neosolaniol, DAS—diacetoxyscirpenol; CV—coefficient of variation of absorbance.

**Table 3 toxins-12-00094-t003:** The absorbance of type B trichothecenes in individual standard solutions.

Mycotoxin	Solvent	Conc., μg/mL	λ_max_/λ_ref._, nm	A (Average, *n* = 3)			CV, %
				Fresh	10 Months	14 Months	
NIV	MeOH	20	217/300	2.36	2.41(+2%)	2.42(+3%)	1.3
	ACN	50	217/300	0.28	0.31(+11%)	0.35(+25%)	11.2
DON	MeOH	50	218/260	0.76	0.82(+8%)	0.81(+7%)	4.0
	ACN	50	218/260	0.99	4.17	2.52	62.1
3-AcDON	MeOH	50	217/300	2.85	2.74(−4%)	2.86(+0.4%)	2.4
	ACN	50	217/300	1.05	1.07(+2%)	0.95(−10%)	6.3
15-AcDON	MeOH	50	221/300	1.08	1.13(+5%)	1.16(+7%)	3.6
	ACN	50	219/300	1.04	1.05(+1%)	1.07(+3%)	1.5
FusX	MeOH	50	216/300	1.22	1.23(+1%)	1.25(+2%)	1.2
	ACN	50	218/300	1.09	1.11(+2%)	--	1.3

MeOH—methanol, ACN—acetonitrile, NIV—nivalenol, DON—deoxynivalenol, 3-Ac, 15-AcDON - 3-acetyl-, 15-acetyl deoxynivalenol, FusX—fusarenone X; CV—coefficient of variation of absorbance.

**Table 4 toxins-12-00094-t004:** The absorbance of zearalenone, its analogues in individual standard solutions.

Mycotoxin	Solvent	Conc., μg/mL	λ_max_/λ_ref._, nm	A (Average, *n* = 3)			CV, %
				Fresh	10 Months	14 Months	
ZEA	ACN	5	235/400	0.95	0.94	0.97	
			273/400	0.40	0.40	0.41	1.4
			314/400	0.18	0.18	0.19	
α-ZEL	ACN	50	235/380	2.75	2.54 (−8%)	2.28 (−17%)	
			272/380	1.29	1.15 (−11%)	1.03 (−20%)	11.3
			315/380	0.59	0.51 (−14%)	0.47 (−20%)	
β-ZEL	ACN	50	239/380	2.97	2.64 (−11%)	2.06 (−30%)	
			274/380	1.34	1.15 (−15%)	0.88 (−34%)	20.6
			315/380	0.58	0.49 (−16%)	0.36 (−38%)	
α-ZAL	ACN	50	218/350	1.20	1.13 (−6%)	0.95 (−21%)	
			264/350	0.59	0.48 (−19%)	0.48 (−19%)	13.3
			302/350	0.24	0.19 (−21%)	0.18 (−25%)	
β-ZAL	ACN	50	218/350	1.89	1.94 (+3%)	1.71 (−10%)	
			261/350	0.87	0.72 (−17%)	0.71 (−18%)	11.7
			301/350	0.35	0.28 (−20%)	0.27(−23%)	

ACN—acetonitrile, ZEA—zearalenone, α-, β-ZEL - α-, β-zearalenol, α-, β-ZAL - α-, β-zearalanol; CV—coefficient of variation of absorbance.

**Table 5 toxins-12-00094-t005:** The absorbance of *Alternaria* toxins in individual standard solutions.

Mycotoxin	Solvent	Conc., μg/mL	λ_max_/λ_ref._, nm	A (Average, *n* = 3)			CV, %
				Fresh	10 Months	14 Months	
AOH	MeOH	20	256/380	3.23	3.21	3.19	
			300/380	0.81	0.81	0.82	0.7
			340/380	0.82	0.81	0.82	
AME	ACN	10	256/380	2.71	2.73	2.77	
			300/380	0.57	0.57	0.60	3.0
			340/380	0.62	0.62	0.64	
ALT	MeOH	20	241/380	2.89	in 4 months*: 2.90	n.a.	
			279/380	1.02	1.03	n.a.	0.7
			320/380	0.57	0.57	n.a	
TE	MeOH	20	206/350	2.13	2.29	2.27	
			282/350	1.32	1.32	1.35	1.3

MeOH—methanol, ACN—acetonitrile, AOH—alternariol, AME—alternariol methyl ether, ALT—altenuene, TE—tentoxin; CV—coefficient of variation of absorbance. *—The total storage period for ALT was four months, thus spectra were obtained immediately after preparation and in four months.

**Table 6 toxins-12-00094-t006:** The absorbance of Enns and BEA individual standard solutions.

Mycotoxin	Solvent	Conc., μg/mL	λ_max_/λ_ref._, nm	A (Average, *n* = 3)			CV, %
				Fresh	10 month	14 month	
EnnA	MeOH	20	206/300	1.03	1.01	1.03	1.1
EnnB	MeOH	20	207/300	0.73	0.74	0.72	1.4
BEA	MeOH	20	206/300	1.94	1.97	1.92	1.3

MeOH—methanol, Enn—enniatin, BEA—beauvericin; CV—coefficient of variation of absorbance.

**Table 7 toxins-12-00094-t007:** The absorbance of MO, MPA, CIT, CTV and CPA in individual standard solutions.

Mycotoxin	Solvent	Conc., μg/mL	λ_max_/λ_ref._, nm	A (Average, *n* = 3)			CV, %
				Fresh	10 Months	14 Months	
MO	MeOH	5	227/245	0.60	0.66 (+10%)	0.68 (+13%)	6.4
			259/245	0.03	0.01	0.02	
CIT	MeOH	20	211/280	2.28	2.37 (+4%)	2.40 (+5%)	
			253/280	0.74	0.72 (−3%)	0.73 (−1%)	1.4
			318/280	0.33	0.32 (−3%)	0.33	
MPA	ACN	250 –storage * 2.5—UV-spec.	214/350	2.45	in 4 months**: 2.44	n.a.	
			249/350	0.4	0.4	n.a.	0
			303/350	0.13	0.13	n.a	
CTV	MeOH	20	233/500	0.70	in 4 months: 0.68	n.a.	
			286/500	1.18	1.17	n.a.	0.6
			385/500	1.66	1.57	n.a.	
CPA	MeOH	20	222/350	2.68	in 4 months: 2.75	n.a.	
			281/350	1.43	1.44	n.a.	0.5

MeOH—methanol, ACN—acetonitrile, MO—moniliformin, CIT—citrinin, MPA—mycophenolic acid, CTV—citreoviridin, CPA—cyclopiazonic acid; CV—coefficient of variation of absorbance. *—MPA exhibits high absorptivity, aliquots of standard solution were diluted 100-fold each time before measurements; **—the total storage period for MPA, CTV and CPA was four months, thus spectra were obtained immediately after preparation and in four months.

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
