# Peer review of "Stability of Mycotoxins in Individual Stock and Multi-Analyte Standard Solutions"

_toxins, 2020, doi:10.3390/toxins12020094_

Round 1

Reviewer 1 Report

This study has evaluated the stability of individual mycotoxin stock solutions and mycotoxin mixture solutions in neat solvents. This is an important topic concerning the long-term storage of mycotoxin standard solutions. The experiments are reasonably designed. This manuscript can be considered for publication.
Minor revisions:
1. In the title, "multi-analyte standard solutions" should be better than "multi-analyte LC-MS/MS standard solutions".
2. Line 420, how did the author determine the sensitivity in their method? (calculating the LOD? How?)

Reviewer 2 Report

The manuscript gives a detailed description of mycotoxin stability for analytical purposes.

In general, the English language, sentence structure and grammar need to be completely revised. Sentences are too long and often, the wrong punctuation and grammar is used. Several examples are provided:

Line 36: "Photodegradation is known for AFLs, thermal stability and food processing impact in decontamination was well reviewed by Milani et al. biodegradation of mycotoxins – by Ben Taheur.  Common factors affecting the stability of standard solution are solvent nature, its ability to degrade analyte and to get evaporated, temperature and UV-irradiation.

The literature appears well covered. Please review the manuscript and insert citations when stating "literature data...".

I suggest more information regarding each mycotoxin be inserted into the manuscript. This will help the reader understand why it is important to test the stability of these specific toxins. Where are they found? Do they implicate human health? How? etc.

I suggest all compounds be written in full the first time they are noted in the text. This is the same for all other abbreviated words e.g. HPLC, LC-MS, CV etc.

Please keep numbering consistent: numbers less than 10 should be written in full (e.g. seven days vs. 7-days (line 99). This is not required for temperature, percentages etc.

Please spell-check the entire manuscript. Some obvious mistakes include "proposingly (line 119), rezults (line 101), aliquotes (line 77) etc.

Please insert citations for the paragraph starting at Line 66.

What do the author's mean by 'valid' (line 85)? Is this in terms of reproducibility? Please explain.

Please explain what 'CV' is in full here. Perhaps explain how the CV is achieved. It can be confusing as all mycotoxins are abbreviated (line 89).

I suggest all numbers are written with their appropriate symbols. For example, write "99% and 96%" instead of "99 and 96%" (line 99). This goes for temperature values as well throughout the manuscript.

Numbers with decimal places should use a "." instead of "," (see line 110: 4, 5 hours instead of 4.5 hours). Please change throughout.

Tables should be stand-alone and therefore, all abbreviations should ideally be placed in footnotes. E.g. MeOH should be written as full as methanol in the footnote. The same goes for the mycotoxin abbreviations etc.

Please provide a citation for line 133 and 183 "literature review". 

Where possible, please avoid using hyphens instead of words. I understand that this allows shortening of sentences, but I found them distracting and the sentences were much more difficult to read. An example is listed below: below:

Line 363: "CIT standard solution is prepared in chloroform, CPA – in methanol, MO – in water 362 [13]. Molar absorptivity for CIT is 16100 at 332 nm, for CPA – 20417 at 284 nm, for MO – 5600 363 L*mol-1*cm-1at 260 nm in the corresponding solvents.

Please review Figure 2. Are these error bars? If so, please update the figure legend. I suggest any corresponding data be available within the supplementary files. Enlarging the figure will also help with the overlay of error bars.

Do the author's need to include mycotoxins/compounds in point 4 of the conclusion? No text is available here.

Finally, please further explain the statement about a matrix effect, and how one would mitigate such a thing (line 433).

Round 2

Reviewer 2 Report

The Author's have addressed all of my comments well. The manuscript has been edited and the English language has been significantly improved. The manuscript is much easier to read now.

Please see the attached PDF with minor corrections.

Author Response

We are grateful for the careful reading of our manuscript.

The language and style were corrected in accordance with your suggestions. Figure 2 was enlarged. Comments on matrix effect were added to the corresponding paragraph (lines 516-519).

Best regards.

This manuscript is a resubmission of an earlier submission. The following is a list of the peer review reports and author responses from that submission.

Round 1

Reviewer 1 Report

The manuscript presents data on the stability of individual and mixed mycotoxin standards under storage conditions. The result reported by the authors will serve as a guide to researchers toward effective storage of mycotoxin standards and obtaining of reliable scientific results.

Comments

Line 188, change “chromatographic sirings” to “chromatographic syringes”

Add reference to the literature review in Table 8.

Author Response

Thank you for your comments.

Line 188, change “chromatographic sirings” to “chromatographic syringes”

Corrected.

Add reference to the literature review in Table 8.

The references were added.

Reviewer 2 Report

line no 83- Aflatoxins and Sterigmatocystein

If possible author can explain about the derivatation of aflatoxins, Its play a vital role in resolution and sensitivity of mycotoxins.

Line no-98- Where is the details about B group loss?

Line no- 120- Did author tried different solvent for increase the absorbance of AFLs?

Line no- 128- if possible author can explain about the derivatation of Type A Trichothecences.

Line no-191- Add the details about long time storage.

Line no- 250- Add the details about long time storage.

Line no-391- Need more clear explanation,how the polyethylene extractive of pipette tips.

Line no- 504- Brief conclusion should be need.

Reviewer 3 Report

This MS aims to determine the stability of some mycotoxins and to compare the results with the available literature. However, the authors are unable to state a direct and testable hypothesis and objectives (line 53-59). The title itself should be revised, is it about method development or is it a  stability study? The authors would stick only the most common food and feed-mycotoxins contaminants such as  AFs, OTA, ZEA, DON, FB1 and 2, patulin, T2 and HT-2. Preparation of mycotoxin in solvents should be carefully considered and revised. In the material and method part, I was unable to find the experimental design …the authors mentioned a lot about the standard and the preparation and they missed the protocol for testing the hypothesis (validation study). Line 449-451 “Selection of solvent and concentration of diluted individual stock solutions intended for preparation of multi-mycotoxin standard was based on a review of the literature concerning physical properties and occurrence of mycotoxins”. The authors should select the solvents and also the concentration based on a certified reference. Instead of listed a table to draw a conclusion, I recommend having direct and useful conclusion that support your research questions. Massive English editing is needed.  

Round 2

Reviewer 3 Report

I would stick with my previous comments. There is a major defect in the experimental design related to the standard preparation and stability study, selection of mycotoxins, selection of matrix, solvents....etc. The revised version has many of deletions, insertions, corrections etc and therefore, it is very hard to follow.